# Non-equilibrium quasiparticles in superconducting circuits: photons vs. phonons

**Gianluigi Catelani[1]⋆ and Denis M. Basko[2]†**

**1** JARA Institute for Quantum Information (PGI-11),
Forschungszentrum Jülich, 52425 Jülich, Germany
**2** Laboratoire de Physique et Modélisation des Milieux Condensés,
Université Grenoble Alpes and CNRS, 25 rue des Martyrs, 38042 Grenoble, France

⋆ g.catelani@fz-juelich.de, † denis.basko@lpmmc.cnrs.fr

## Abstract

We study the effect of non-equilibrium quasiparticles on the operation of a superconducting device (a qubit or a resonator), including heating of the quasiparticles by the device operation. Focusing on the competition between heating via low-frequency photon absorption and cooling via photon and phonon emission, we obtain a remarkably simple non-thermal stationary solution of the kinetic equation for the quasiparticle distribution function. We estimate the influence of quasiparticles on relaxation and excitation rates for transmon qubits, and relate our findings to recent experiments.

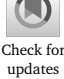
# 1 Introduction

Superconducting devices have been under development for several years for applications in detection [1], nonlinear microwave amplification [2–4], quantum information processing [5], and quantum metrology [6,7]. Intrinsic excitations in the superconductor, known as Bogoliubov quasiparticles, can be detrimental to such devices, for example limiting the sensitivity of detectors and causing decoherence in qubits. The devices are routinely operated at temperatures so low that no quasiparticles should be present in thermal equilibrium. However, a significant number of residual non-equilibrium quasiparticles is typically detected, and it is well established that their density (normalized to the Cooper pair density) can be $x_{\mathrm{qp}} \sim 10^{-5}$–$10^{-8}$ [8–10]. Much less is known about the energy distribution of these quasiparticles.

Many experiments involving residual quasiparticles in qubits [11–13] are successfully described by the theory [14,15], based on the assumption of a fixed average quasiparticle distribution which perturbs the device operation; the resulting net effect of the quasiparticles is equivalent to that of a frequency-dependent resistance included in the circuit. The fixed distribution assumption is valid in the weak signal regime, when the back-action of the superconducting condensate excitations (resonator photons or qubit excitations, hereafter all referred to as photons for the sake of brevity) on the quasiparticles can be neglected, or for studying observables which are not sensitive to the quasiparticle energy distribution function, but only to the quasiparticle density. This assumption must be reconsidered when the probing signal is strong enough to modify the quasiparticle distribution and the latter can affect the quantities which are measured.

Some efforts in this direction have been made. In the numerical work [16], the external circuit was treated classically, so it tended to heat the quasiparticles to infinite temperature, and the distribution was stabilized by phonon emission only (see also [17] and references therein for related experiments). In [18], the quasiparticle distribution was assumed to be entirely determined by the external circuit (both heating by the drive and cooling by photon emission were included at the quantum level), while phonon emission was neglected.

Here we study analytically the competition between quasiparticle heating via absorption of low-energy photons (*i.e.*, with energy $\omega_0 \ll 2\Delta$, twice the superconducting gap; we use units with $\hbar = 1$, $k_B = 1$ throughout the paper) and cooling via both photon emission in the external circuit and phonon emission in the material, as schematically shown in Fig. 1. We obtain a nonthermal stationary solution of the kinetic equation for the quasiparticle distribution function. Using our solution we estimate the influence of non-equilibrium quasiparticles on relaxation and excitation rates for transmon qubits, and relate our findings to recent experiments.

The paper is organized as follows: we first introduce the kinetic equation and discuss the main properties of its solution for different regimes (conditions for applicability of the kinetic equation and detailed derivation of the solution are given in appendices). We then discuss our results in light of recent experiments with superconducting qubits.

# 2 Kinetic equation

The quasiparticle distribution function $f(\epsilon)$ satisfies the kinetic equation that we write as

$$\frac{\partial f(\epsilon)}{\partial t} = \mathrm{St}_{\mathrm{t}} f(\epsilon) + \mathrm{St}_{\mathrm{n}} f(\epsilon), \tag{1}$$

where we measure the energy $\epsilon$ from the superconducting gap and assume $0 < \epsilon \ll \Delta$. The two terms on the right-hand side represent collision integrals due to absorption/emission of

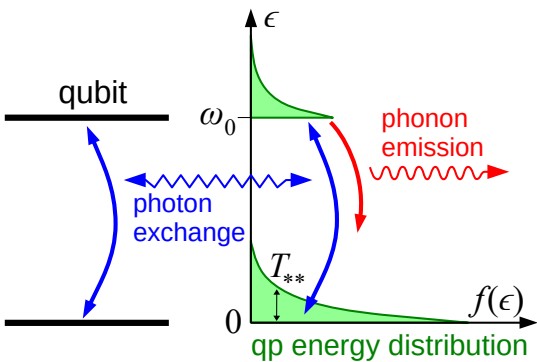

Figure 1: A schematic representation of the model under consideration. The quasi-particle distribution function $f(\epsilon)$ (shaded areas) is affected by photon exchange with *e.g.* a qubit and by the emission of phonons. As explained in the text, for "cold" quasiparticles the width $T_{**}$ of the distribution function [Eq. (11)] is small compared to the photon frequency $\omega_0$, and the distribution function has a second, smaller peak at that frequency (replicas at higher multiples of $\omega_0$ can be neglected in the cold quasiparticle limit).

phoTons and phoNons, respectively. The photon term can be written as [18]

$$\text{St}_\text{t} f(\epsilon) = \bar{n}\Gamma_0 \sqrt{\frac{\omega_0}{\epsilon - \omega_0}}\Big[ f(\epsilon - \omega_0) - e^{\omega_0/T_0} f(\epsilon) \Big] +$$
$$+ \bar{n}\Gamma_0 \sqrt{\frac{\omega_0}{\epsilon + \omega_0}}\Big[ e^{\omega_0/T_0} f(\epsilon + \omega_0) - f(\epsilon) \Big]. \tag{2}$$

Here the square roots approximate the density of states near the gap edge, $\epsilon \ll \Delta$; for $\epsilon < \omega_0$ the first term should be set to zero. $\bar{n}$ is the average number of excitations in the superconducting subsystem with transition frequency $\omega_0$; for a qubit $0 \leq \bar{n} < 1$, while for a harmonic system such as a resonator any $\bar{n} \geq 0$ is allowed; this is in contrast to the classical approach of Ref. [16], valid only for $\bar{n} \gg 1$. The effective temperature $T_0$ is defined by the relation $e^{\omega_0/T_0} = (1 \mp \bar{n})/\bar{n}$, with the upper (lower) sign for a two-level qubit (harmonic) subsystem. $\Gamma_0$ characterizes the rate of photon absorption/emission by the quasiparticle. For a weakly-anharmonic, single-junction qubit such as the transmon it is given by

$$\Gamma_0 = \frac{\delta}{2\pi\Delta} \sqrt{\frac{\omega_0 \Delta}{2}}, \tag{3}$$

with $\delta$ being the mean level spacing of the superconducting islands. For a resonator, one should use the mean level spacing in the whole resonator volume, occupied by the quasiparticles, while $2\pi\Delta$ should be replaced by $Q\omega_0$ with $Q$ being the resonator quality factor if the material resistivity were the same as in the normal state.

We do not address photon dynamics here, assuming the photon state to be stationary and fixed by the external circuit. In principle, one can write coupled equations for the photon density matrix and $f(\epsilon)$, as in Ref. [18]; their solution in the presence of phonons remains an open question. Also, we neglect multiphoton absorption/emission and assume the photon system to be either strictly two-level or strictly harmonic; then each act of absorbtion/emission involves only energy $\omega_0$, and the photon state enters only via $\bar{n}$. Going beyond these assumptions would allow quasiparticles to absorb/emit multiples of $\omega_0$ or energies slightly different from $\omega_0$ due to weak anharmonicity in the resonator or broadening of the qubit transition (we

remind that the ratio between anharmonicity and broadening determines if the photon system should be treated as two-level or harmonic [18]).

We assume the phonons to be kept at constant low temperature $T_{\mathrm{ph}} \ll \omega_0$. Then, the quasiparticle-phonon scattering collision integral can be written as [19–21]:

$$\mathrm{St_n} f(\epsilon) = \int_0^\infty \frac{2\pi F(\omega)\, d\omega}{1 - e^{-\omega/T_{\mathrm{ph}}}} \frac{2\epsilon + \omega}{\sqrt{2\Delta(\epsilon + \omega)}} \left[ f(\epsilon + \omega) - e^{-\omega/T_{\mathrm{ph}}} f(\epsilon) \right] +$$

$$+ \int_0^\epsilon \frac{2\pi F(\omega)\, d\omega}{1 - e^{-\omega/T_{\mathrm{ph}}}} \frac{2\epsilon - \omega}{\sqrt{2\Delta(\epsilon - \omega)}} \left[ e^{-\omega/T_{\mathrm{ph}}} f(\epsilon - \omega) - f(\epsilon) \right], \tag{4}$$

where we assume the quasiparticles to be non-degenerate, $f(\epsilon) \ll 1$, so all terms quadratic in $f(\epsilon)$ (in particular, quasiparticle recombination) are neglected. This approximation is discussed in detail in Appendix A. The function $F(\omega)$ is defined as

$$F(\omega) \equiv \frac{\Xi^2 \omega^2}{8\pi^2 \hbar v_F \rho_0 v_s^4} \equiv \frac{\hbar^3 \Sigma \omega^2}{48\pi \zeta(5)\, \nu_{\mathrm{n}}}. \tag{5}$$

Here $\zeta(x)$ is the Riemann zeta function, $\Xi$ is the deformation potential, $v_F$ and $v_s$ are the Fermi velocity and the speed of sound, $\rho_0$ is the mass density of the material, $\nu_{\mathrm{n}}$ is the density of states at the Fermi level for the material in the normal state, taken per unit volume and for both spin projections. These material parameters are conveniently wrapped into the coefficient $\Sigma$, which controls energy exchange between electrons and phonons for the material in the normal state: the power per unit volume transferred from electrons to phonons, kept at temperatures $T_{\mathrm{e}}$ and $T_{\mathrm{ph}}$, respectively, is given by $\Sigma(T_{\mathrm{e}}^5 - T_{\mathrm{ph}}^5)$ [22]. This relationship as well as Eq. (5) are appropriate for a clean metal, when the electron elastic mean free path due to static impurities is longer than the mean free path due to the electron-phonon scattering. In the opposite limit, $F(\omega)$ is proportional to a different power of $\omega$: $F(\omega) \propto \omega$ for impurities with fixed positions [23] and $F(\omega) \propto \omega^3$ for impurities which move together with the phonon lattice deformation [24, 25]. Although at low energies, discussed here, the system is expected to be in the diffusive limit, the clean-limit expressions usually agree better with experiments (see Ref. [26] for a review). Thus, here we use the clean-limit formula; the diffusive limit is discussed in Appendix D where we show that the results are qualitatively similar.

We assume the phonon temperature to be very low, much smaller than the typical quasiparticle energy. Then the last term in Eq. (4) determines the quasiparticle relaxation rate by phonon emission [1]:

$$\Gamma_{\mathrm{ph}}(\epsilon) = \frac{16}{315\,\zeta(5)} \frac{\Sigma \epsilon^{7/2}}{\nu_{\mathrm{n}} \sqrt{2\Delta}} \equiv \frac{1}{\tau_{\mathrm{ph}}} \left( \frac{\epsilon}{\Delta} \right)^{7/2}. \tag{6}$$

When we neglect the absorption of phonons (i.e., setting $T_{\mathrm{ph}} = 0$), the phonon collision integral simplifies:

$$\mathrm{St_n} f(\epsilon) = \frac{105}{128} \int_\epsilon^\infty \frac{f(\epsilon')}{\tau_{\mathrm{ph}}} \frac{(\epsilon' + \epsilon)(\epsilon' - \epsilon)^2 d\epsilon'}{\sqrt{\Delta^7 \epsilon'}} - \left( \frac{\epsilon}{\Delta} \right)^{\frac{7}{2}} \frac{f(\epsilon)}{\tau_{\mathrm{ph}}}. \tag{7}$$

Some consequences of $T_{\mathrm{ph}} > 0$ are explored in Appendix C. From now on, with kinetic equation we will mean Eq. (1) with the right hand side given by the sum of this simplified expression plus Eq. (2). In the steady state, $\partial f / \partial t = 0$, the kinetic equation reduces to an integral equation for $f(\epsilon)$. In the next section we study the solution of this equation and we identify two main regimes (see Fig. 2).

---

[1]To avoid confusion, we note that the symbol $\tau_{\mathrm{ph}}$ used here corresponds to $\tau_{\mathrm{ph}}(\Delta)$ of Ref. [18]. It is related to $\tau_0$ of Ref. [19] by $\tau_{\mathrm{ph}} = \tau_0 \sqrt{2}(105/128)(T_c/\Delta)^3 \simeq 0.21\tau_0$, where $T_c$ is the superconducting critical temperature.

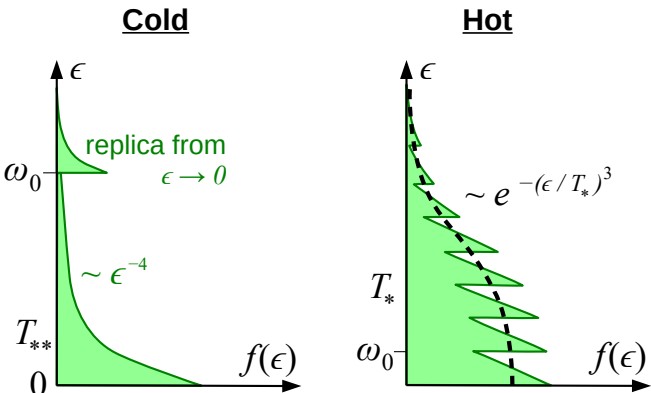

Figure 2: A schematic representation of the quasiparticle distribution function $f(\epsilon)$ (green shaded areas) in the two regimes, cold and hot (left and right panels), described in Secs. 3.1 and 3.2, respectively. For cold quasiparticles, most population is concentrated at energies $\epsilon \lesssim T_{**} \ll \omega_0$; above $T_{**}$, the distribution function decays as $f(\epsilon) \sim \epsilon^{-4}$; $f(\epsilon)$ has a second, smaller peak at $\epsilon = \omega_0$ (replicas at higher multiples of $\omega_0$ can be neglected). In the hot quasiparticle regime, $f(\epsilon)$ has a sawtooth dependence with the period $\omega_0$, on top of a smooth envelope (black dashed curve), characterized by a scale $\epsilon \sim T_* \gg \omega_0$. (Note the different scales for $\omega_0$ in the two panels.)

## 3 Solution: two main regimes

### 3.1 Cold quasiparticle regime

The first regime occurs when there are few excitations to absorb and/or the electron-phonon relaxation is sufficiently strong, so the quasiparticles are more likely found at low energies, $\epsilon \ll \omega_0$. From time to time, a quasiparticle absorbs a quantum $\omega_0$ and is promoted to energies above $\omega_0$, and then quickly relaxes back to lower energies by emitting either a quantum $\omega_0$, or a phonon. Thus, we can find the distribution at energies just above $\omega_0$ by perturbation theory from the stationary kinetic equation. At those energies, we can neglect the first term in the right hand side of Eq. (7) because the $\epsilon'$ integral is dominated by $\epsilon' - \epsilon \ll \omega_0$, while in the second term $\epsilon \approx \omega_0$ (the smallness of the first term can be checked self-consistently after the solution is found). Then, neglecting also the occupation at energies larger than $2\omega_0$, we obtain:

$$f(\epsilon + \omega_0) \approx \frac{f(\epsilon)}{e^{\omega_0/T_0} + \sqrt{\epsilon/\omega_0}\,\Lambda}, \tag{8}$$

where we introduced a dimensionless parameter

$$\Lambda \equiv \frac{(\omega_0/\Delta)^{7/2}}{\bar{n}\Gamma_0 \tau_{\text{ph}}}. \tag{9}$$

The condition $e^{\omega_0/T_0} \gg 1$ or $\Lambda \gg 1$ ensures that the perturbative approach is valid and thus defines the cold quasiparticle regime.[2]

---

[2]One may worry that if the condition $T_0 \ll \omega_0$ is not satisfied, $f(\epsilon + \omega_0) \approx f(\epsilon)$ for small $\epsilon$, so the perturbation theory does not work. Note, however, that if we try to find $f(\epsilon + 2\omega_0)$ from the stationary kinetic equation focusing

Substituting $f(\epsilon + \omega_0)$ from Eq. (8) into the kinetic equation at low energies $\epsilon \ll \omega_0$, we obtain

$$\int_\epsilon^{\omega_0} \frac{(\epsilon' + \epsilon)(\epsilon' - \epsilon)^2 f(\epsilon') d\epsilon'}{\sqrt{\omega_0^7 \epsilon'}} + \int_0^{\omega_0} \frac{f(\epsilon') d\epsilon'/\omega_0}{e^{\omega_0/T_0} + \sqrt{\epsilon'/\omega_0}\,\Lambda} =$$

$$= \frac{128}{105}\left[\frac{\sqrt{\epsilon/\omega_0}}{e^{\omega_0/T_0} + \sqrt{\epsilon/\omega_0}\,\Lambda} + \left(\frac{\epsilon}{\omega_0}\right)^{7/2}\right] f(\epsilon). \tag{10}$$

On the right-hand side, the first term in the brackets represents the out-scattering to the higher levels at $\epsilon > \omega_0$ (minus the photon emission contribution), while the second term is due to phonon emission, when the quasiparticle is transferred to lower energies. This second term dominates at energies

$$\epsilon \gg T_{**} \sim \min\left\{\omega_0 \Lambda^{-2/7},\, \omega_0 e^{-\omega_0/(3T_0)}\right\}, \tag{11}$$

and the cold quasiparticle condition can be expressed as $T_{**} \ll \omega_0$. The last term on the left-hand side of Eq. (10) represents the uniform incoming flux from the energies between $\omega_0$ and $2\omega_0$ (higher energies are neglected, as discussed above). This term is smaller than the first one (incoming flux from energies below $\omega_0$) as long as $\epsilon \ll \omega_0$ (see Appendix B). In both integrals, we can push the upper integration limit to infinity if the integrand decreases quickly enough. Then, we are left with the equation

$$\int_\epsilon^\infty \frac{(\epsilon' + \epsilon)(\epsilon' - \epsilon)^2 f(\epsilon') d\epsilon'}{\sqrt{\epsilon'}} = \frac{128}{105} \epsilon^{7/2} f(\epsilon), \tag{12}$$

which has a remarkably simple exact solution $f(\epsilon) = C/\epsilon^4$ with an arbitrary constant $C$. This power-law form justifies the change in integration limit and is valid for energies $T_{**} \ll \epsilon \ll \omega_0$; it resembles the Kolmogorov spectrum of turbulence [27], and describes a similar physical situation: the flow of probability, which is injected at high energies, $\epsilon \gtrsim \omega_0$, and flows to lower energies, down to $\epsilon \sim T_{**}$ where it encounters an effective sink, represented by the first term on the right-hand side of Eq. (10). In our case, the probability is reinjected back at higher energies by absorbing a quantum $\omega_0$.

The behavior of the solution at energies $\epsilon \lesssim T_{**}$ depends on the relation between the two dimensionless parameters, $e^{\omega_0/T_0}$ and $\Lambda$. While we are unable to find an analytical solution in this region, it is possible to establish the qualitative character of the solution. Let us drop the second term on the left-hand side of Eq. (10) and introduce new variables

$$x = \frac{\epsilon}{\omega_0}\left(\frac{\Lambda}{e^{\omega_0/T_0}}\right)^2, \quad f(\epsilon) = \mathcal{F}_r(x)\left(\frac{r\sqrt{x}}{1 + \sqrt{x}} + x^{7/2}\right)^{-1}, \tag{13}$$

where $r = \Lambda^6/e^{7\omega_0/T_0}$. This gives an equation for $\mathcal{F}_r(x)$:

$$\mathcal{F}_r(x) = \frac{105}{128}\int_x^\infty \frac{(x + y)(x - y)^2}{r(1 + \sqrt{y})^{-1} + y^3} \frac{\mathcal{F}_r(y)\,dy}{y}. \tag{14}$$

---

on energies close to $2\omega_0$ and neglecting $f(\epsilon + 3\omega_0)$ as well as the first term in Eq. (7), we obtain

$$f(\epsilon + 2\omega_0) \approx \frac{f(\epsilon + \omega_0)}{e^{\omega_0/T_0} + 1/\sqrt{3} + \sqrt{128}\,\Lambda},$$

so having just one of the conditions $T_0 \ll \omega_0$ or $T_* \ll \omega_0$ suffices to neglect $f(\epsilon + 2\omega_0)$ and higher.

The $1/\epsilon^4$ asymptotics of $f(\epsilon)$ translates into $\mathcal{F}_r(x) \sim 1/\sqrt{x}$ at $x \gg x_{**} \approx \min\{r^{2/7}, r^{1/3}\}$, so the integral converges at the upper limit. $\mathcal{F}_r(0)$ is finite, since at low energies the integral is also well-behaved. The contribution of $y \ll x_{**}$ is suppressed by the numerator, so $\mathcal{F}_r(0)$ and $\mathcal{F}_r(x_{**})$ are of the same order. This qualitative analysis is supported by numerical findings which show that to good accuracy the function $\mathcal{F}_r(x)$, upon rescaling $x$ by $x_{**}$, takes a form independent of $r$ (Appendix B).

Note that because of the square root divergence of $f(\epsilon)$ as $\epsilon \to 0$, the normalization coefficient depends weakly (logarithmically) on the phonon temperature when $T_{\mathrm{ph}} \ll T_{**}$. On the other hand, Eq. (8) and the power-law decay at large energy remain valid even when $T_{\mathrm{ph}} > T_{**}$, although the latter starts from an energy large compared to $T_{\mathrm{ph}}$, as discussed in Appendix C.

## 3.2 Hot quasiparticle regime

If $\bar{n}$ (and hence $T_0$) is sufficiently large, the width $T_{**}$ of the distribution function may exceed $\omega_0$. When both $\omega_0/T_0$, $\Lambda \ll 1$, the quasiparticles can absorb many excitations and their typical energy becomes large compared to $\omega_0$, which defines the hot quasiparticle regime. Still, this does not mean that $f(\epsilon)$ becomes smooth on the scale $\epsilon \sim \omega_0$: the singularity in the density of states at $\epsilon \to 0$ is imprinted in $f(\epsilon \to \omega_0)$, $f(\epsilon \to 2\omega_0)$, etc., by photon absorption, giving rise to a series of peaks in $f(\epsilon)$ at integer multiples of $\omega_0$ (cf. Fig. 2), which were observed in the numerical results of Ref. [16]. Below we focus on large-scale features of the distribution function, its fine structure lying beyond the scope of the present paper. Then, if $f(\epsilon)$ is understood as the smooth envelope, we can approximate the photon collision integral in Eq. (2) by a diffusion operator:

$$\mathrm{St}_{\mathrm{t}} f(\epsilon) \simeq \bar{n}\Gamma_0 \sqrt{\omega_0^5 \epsilon}\, \frac{\partial}{\partial\epsilon} \left\{ \frac{e^{-\epsilon/T_0}}{\epsilon} \frac{\partial}{\partial\epsilon} \left[ e^{\epsilon/T_0} f(\epsilon) \right] \right\}, \tag{15}$$

whose form ensures the correct steady-state solution $f \sim e^{-\epsilon/T_0}$ when phonon emission is neglected [18]. In this approximation, it is convenient to introduce the temperature scale

$$T_* \equiv \left[ (\bar{n}\Gamma_0\tau_{\mathrm{ph}})^2 \omega_0^5 \Delta^7 \right]^{1/12} = \omega_0 \Lambda^{-1/6}. \tag{16}$$

For the quasiparticle to be hot, we need $T_0$, $T_* \gg \omega_0$.

Let us first assume $\omega_0 \ll T_* \ll T_0$ (more precisely, we are considering the limits $\omega_0 \to 0$, $T_0 \to \infty$, while keeping $T_* = \mathrm{const}$). Then we can replace the exponential terms in the curly brackets of Eq. (15) with unity and introduce dimensionless variables $x = \epsilon/T_*$ and $y = \epsilon'/T_*$. As a result, the steady-state equation acquires a parameter-free form:

$$0 = \sqrt{x}\, \partial_x \left[ x^{-1} \partial_x f(x) \right] - x^{7/2} f(x) + \frac{105}{128} \int_x^\infty \frac{dy}{\sqrt{y}} (x+y)(x-y)^2 f(y). \tag{17}$$

We can find approximate solutions to Eq. (17) in the low ($x \ll 1$) and high ($x \gg 1$) energy limits; the solutions will contain unknown coefficients, which we will fix by solving the equation numerically. In the high-energy regime, we can drop the last term, and with the change of variable $x = 4^{1/6}\sqrt{z}$ we obtain the Airy equation:

$$0 = \partial_z^2 f - z f. \tag{18}$$

Thus, the high-energy part of the solution is expressed in terms of the Airy function:

$$f(x \gg 1) \approx f_\infty \mathrm{Ai}\big(x^2/4^{1/3}\big), \tag{19}$$

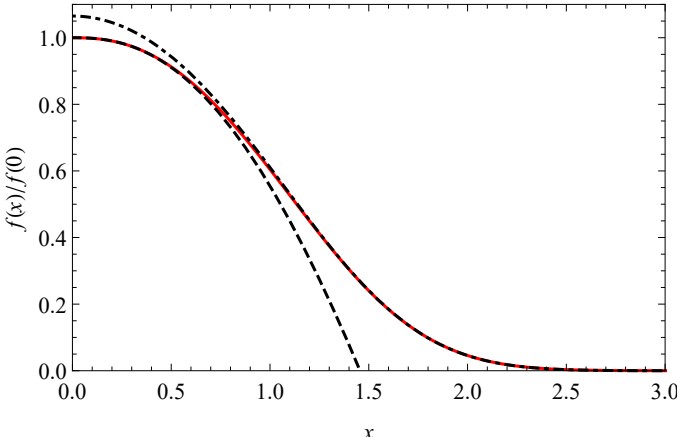

Figure 3: Red solid line: numerical solution to Eq. (17). Black dashed: low energy approximation, Eq. (20). Black dot-dashed: high energy approximation, Eq. (19).

with some coefficient $f_\infty$. At low energies $x \to 0$, in Eq. (17) we can drop the term proportional to $x^{7/2}$ and set the lower integration limit to zero. We can then find a solution in the form of an expansion:

$$f(x) \approx f_0 \left(1 - ax^{5/2} + bx^{7/2} + \dots\right), \tag{20}$$

with

$$a = \frac{21}{32} \int_0^\infty dy \, y^{5/2} \frac{f(y)}{f_0}, \quad b = \frac{5}{32} \int_0^\infty dy \, y^{3/2} \frac{f(y)}{f_0}, \tag{21}$$

and some coefficient $f_0$, whose relation to $f_\infty$ is in principle determined by matching the solutions at $x \sim 1$; here we find the relation accurately by comparing to numerical result.

We solve Eq. (17) numerically as follows: we discretize variables $x$ and $y$, so that the equation becomes a matrix equation and we find the vector with the smallest eigenvalue in absolute value. We repeat this process on finer grids and then extrapolate to zero spacing; in the extrapolation, the smallest eigenvalue tends to zero. By fitting the numerical solution with Eq. (19) we find $f_\infty/f_0 \simeq 3.00$, while using the definitions in Eq. (21), numerical integration gives $a \simeq 0.564$ and $b \simeq 0.119$. With these values both the high- and low-energy formulas, Eqs. (19) and (20), fit well the numerical solution, see Fig 3. Finally, to find the proper normalization, we can use the numerical result

$$\int_0^\infty \frac{dx}{\sqrt{x}} f(x) \simeq 2.1 f_0. \tag{22}$$

Let us briefly discuss the opposite limit, $T_* \gg T_0 \gg \omega_0$. In this regime we can use Eq. (15), but not the simpler form that led to the first term in the right hand side of Eq. (17). In this limit, the bulk of the distribution is $f(\epsilon) \propto e^{-\epsilon/T_0}$, and only at very high energies, $\epsilon \gtrsim T_\infty \equiv \sqrt{T_*^3/(2T_0)}$, the tail is suppressed by the phonon emission:

$$f(\epsilon) \propto \begin{cases} e^{-\epsilon/T_0}, & \epsilon \ll T_\infty, \\ e^{-\epsilon/2T_0 - \epsilon^2/2\sqrt{2}T_0 T_\infty}, & |\epsilon - T_\infty| \ll T_\infty, \\ e^{-\epsilon/2T_0} \mathrm{Ai}\left(\epsilon^2/4^{1/3} T_*^2\right), & \epsilon \gg T_\infty. \end{cases} \tag{23}$$

We see that when $T_0 \ll T_*$, the Boltzmann-like exponential decay $f(\epsilon) \sim e^{-\epsilon/T_0}$ is valid up to the very high energy $T_* \sqrt{T_*/2T_0}$, above which the distribution function decays faster than exponentially. On the other hand, when $T_0 \gg T_*$, the quasiparticle distribution function is governed by $T_*$ rather than $T_0$: it changes little for $\epsilon \lesssim T_*$ and decays faster than exponentially for $\epsilon > T_*$.

# 4 Discussion: relevance to experiments

The above results for the quasiparticle distribution function are valid both for a harmonic and a two-level (qubit) superconducting system, if the appropriate relationship between $\bar{n}$ and $T_0$ is used. We focus henceforth on the qubit case to investigate to what extent a qubit can heat the quasiparticles. Since $T_0/\omega_0 = 1/\ln(1/\bar{n} - 1)$, the condition $T_0 > \omega_0$ translates into $\bar{n} > 1/(1+e) \approx 0.27$. Noting that the excited state occupation $\bar{n}$ of an undriven qubit is generally below 0.3 (in most cases being between a few percent and not much above 10%, with some qubits being much colder, $\bar{n} \sim 0.1\%$ [28]), we conclude that an undriven qubit cannot make the quasiparticles hot. For a strongly driven qubit, we can have $\bar{n} \to 1/2^-$ and hence $T_0 \to \infty$ [18]. In this limit, we should check if $T_* \gg \omega_0$. The largest values of $T_*/\omega_0$ are reached for low-frequency qubits made of small islands, since the rate $\Gamma_0$ is inversely proportional to the volume. Considering a typical aluminum island of volume $\sim 0.02\,\mu\mathrm{m}^3$, we have $\Gamma_0 \sim 10^5\,\mathrm{Hz}$ [18]. For the time $\tau_{\mathrm{ph}}$, we use the experimental results for thin films in Refs. [29] and [30] to estimate $\tau_{\mathrm{ph}} \sim 10\,\mathrm{ns}$. Then for typical qubit frequencies $\omega_0 = 4$ to $8\,\mathrm{GHz}$, we find that $T_*/\omega_0$ varies between 1.2 and 0.8. Hence we conclude that even under strong driving a qubit cannot significantly heat the quasiparticles to energies much above its frequency. This conclusion is not too sensitive to the used parameters: for instance, increasing $\tau_{\mathrm{ph}}$ by two orders of magnitude only doubles the calculated $T_*$ because of the very weak power $\Lambda^{-1/6}$ in Eq. (16).

We can extend the considerations above to include the possibility that the quasiparticles are not heated directly by the qubit, but indirectly due to the interaction of the qubit with another mode. This mode could be a resonator mode, or a spurious (harmonic) mode of the chip. For instance, take a resonator, coupled to the qubit with coupling strength $g$; typically, the coupling strength is of order 100 MHz and the resonator-qubit detuning $\delta\omega$ is of order 1 GHz. This implies that the quasiparticle rate $\Gamma_0$ is suppressed by about two orders of magnitude, $\Gamma_0 \to \Gamma_0(g/\delta\omega)^2$. Therefore, even for a small qubit, we do not expect significant heating unless the mode is populated with hundreds of photons.

Let us now consider an undriven qubit; as discussed above, for the experimentally observed range of $\bar{n}$ the quasiparticles are cold. However, for small-island qubits the distribution function width $T_{**}$ is given by $\omega_0 e^{-\omega_0/(3T_0)}$, while for larger qubits (3D transmon [9, 11], Xmon [31]) with electrode volume of order $10^3$–$10^4\,\mu\mathrm{m}^3$, it is given by $\omega_0\Lambda^{-2/7}$, see Eq. (11). The different regimes for $T_{**}$ lead to different behaviors for the quasiparticle-induced excitation rate. In both cases, since $T_{**} \ll \omega_0$ the relaxation rate is approximately given by [15]

$$\Gamma_{10}^{\mathrm{qp}} = \frac{2\omega_0}{\pi}\int_0^\infty \frac{f(\epsilon)}{\sqrt{\epsilon(\epsilon+\omega_0)}}\,d\epsilon \approx \frac{\omega_0}{\pi}\sqrt{\frac{2\Delta}{\omega_0}}\,x_{\mathrm{qp}}, \tag{24}$$

where

$$x_{\mathrm{qp}} = \int_0^\infty f(\epsilon)\sqrt{\frac{2\Delta}{\epsilon}}\,\frac{d\epsilon}{\Delta} = \frac{n_{\mathrm{qp}}}{\nu_{\mathrm{n}}\Delta} \tag{25}$$

is the quasiparticle volume density $n_{\mathrm{qp}}$ normalized by the Cooper pair density. For the excitation rate, we use Eq. (8) to write

$$\Gamma_{01}^{\mathrm{qp}} = \frac{2\omega_0}{\pi}\int_0^\infty \frac{f(\epsilon+\omega_0)}{\sqrt{\epsilon(\epsilon+\omega_0)}}\,d\epsilon \approx \frac{2\omega_0}{\pi}\int_0^\infty \frac{f(\epsilon)}{\sqrt{\epsilon(\epsilon+\omega_0)}}\frac{d\epsilon}{e^{\omega_0/T_0}+\sqrt{\epsilon/\omega_0}\Lambda}, \tag{26}$$

with the integral dominated by contributions from energies $\epsilon \lesssim T_{**}$ For small qubits, $e^{-\omega_0/(3T_0)} < \Lambda^{-2/7}$, in the relevant energy range the last denomi-

nator is approximately $e^{\omega_0/T_0}$ and therefore

$$\frac{\Gamma_{01}^{\text{qp}}}{\Gamma_{10}^{\text{qp}}} \approx e^{-\omega_0/T_0} = \frac{\bar{n}}{1-\bar{n}} \; . \tag{27}$$

This ratio has the detailed balance form; note, however, that it is determined by the qubit occupation rather than the energy scale of the quasiparticles, which are not in thermal equilibrium. For large qubits, $e^{-\omega_0/(3T_0)} > \Lambda^{-2/7}$, we cannot neglect the term proportional to $\Lambda$ in the denominator, and therefore we conclude $\Gamma_{01}^{\text{qp}}/\Gamma_{10}^{\text{qp}} \ll \bar{n}/(1-\bar{n})$. We should point out that for large qubits we estimate $T_{**} \lesssim T_{\text{ph}}$: not surprisingly, in a large device the quasiparticles should be close to thermal equilibrium with the phonon bath. However, the found inequality for the ratio of quasiparticle rates still holds, since it is based on the use of Eq. (8). This inequality, together with Eq. (24), validates the use of the density $x_{\text{qp}}$ as the relevant dynamical variable affecting qubit relaxation. The dynamics of $x_{\text{qp}}$ in the presence of traps has been studied in recent works [32, 33], where possible ways to improve qubits performance are analyzed.

Some recent experiments [12, 34] with large qubits report that at low temperatures the main relaxation mechanism is a "residual" (non-quasiparticle) one, $\Gamma_{10}^{\text{r}} \gtrsim \Gamma_{10}^{\text{qp}}$. Then, if we assume that the qubit is the main source of quasiparticle non-equilibrium, the above considerations imply $\Gamma_{01}^{\text{qp}}/\Gamma_{10}^{\text{qp}} \ll \bar{n} \approx (\Gamma_{01}^{\text{qp}} + \Gamma_{01}^{\text{r}})/\Gamma_{10}^{\text{r}}$. This inequality can be satisfied only if $\Gamma_{01}^{\text{qp}} \ll \Gamma_{01}^{\text{r}}$, whereas experiments indicate that the opposite inequality holds [28, 34]. This discrepancy could indicate that one should account for a (currently unknown) energy-dependent source of pair breaking photons and/or phonons generating hot quasiparticles; in other words, the source of the quasiparticles should be identified, possibly via experiments with resonators [35]. Alternatively, a more detailed modeling of the systems under investigation might be necessary, for example by including details about the geometry of the superconducting electrodes or properties of the substrate and interfaces that have been neglected so far.

# 5 Conclusion

In summary, we have studied the distribution function for quasiparticles interacting with photons of frequency $\omega_0$ and a low-temperature phonon bath. We have identified the dimensionless parameters that control whether the quasiparticles are cold (i.e., mostly occupying states with energy below $\omega_0$) or hot, see Eqs. (9) and (16) and the text following them; those parameters are determined by the average number of photon excitations $\bar{n}$, the product $\Gamma_0 \tau_{\text{ph}}$ characterizing the relative strengths of quasiparticle coupling to photons and phonons, and the ratio between photon frequency and superconducting gap $\Delta$. Applying our results to transmon qubits, we conclude that the qubit itself cannot overheat the quasiparticles. The extension of our findings to other systems, such as resonators and multi-junction circuits (arrays [36, 37], fluxonium qubit [13]), will be presented elsewhere.

# Acknowledgements

We acknowledge discussions with M. Devoret, L. Glazman, M. Houzet, H. Moseley, K. Serniak, and R. Schoelkopf. This work was supported in part by the Internationalization Fund - Seed Money initiative of Forschungszentrum Jülich.

## A  Neglected terms in the kinetic equation

Besides quasiparticle-phonon scattering, phonon absorption can result in production of a pair of quasiparticles, and a pair of quasiparticles can recombine by emitting a phonon. In the absence of extrinsic processes, the balance between these two processes determines the quasiparticle density (that is, the normalization of the distribution function), since the scattering processes considered in the main text redistribute the quasiparticle energy but do not change their number. However, if the generation/recombination rates are small compared to the rates of the scattering processes, the latter determine the shape of the distribution function. The generation rate is proportional to $e^{-2\Delta/T_{\mathrm{ph}}}$ and is therefore negligibly small. To estimate the importance of recombination, we note that it can be included in the kinetic equation (1) by adding a term

$$\mathrm{St}_{\mathrm{rec}} f(\epsilon) = -4\pi F(2\Delta) \int_0^\infty \sqrt{\frac{\Delta}{2\epsilon'}} f(\epsilon) f(\epsilon')\, d\epsilon' \equiv -\Gamma_{\mathrm{rec}} f(\epsilon), \tag{28}$$

where the recombination rate can be expressed using Eq. (25):

$$\Gamma_{\mathrm{rec}} = \pi F(2\Delta) \frac{2 n_{\mathrm{qp}}}{\nu_{\mathrm{n}}} = \frac{315\sqrt{2}}{96} \frac{x_{\mathrm{qp}}}{\tau_{\mathrm{ph}}}. \tag{29}$$

The combination $2 n_{\mathrm{qp}}/\nu_{\mathrm{n}}$ has the meaning of the normal-state level spacing in a volume occupied by one quasiparticle.

For quasiparticles excited by photons to energy of order $\omega_0$, we can consider the condition $\Gamma_{\mathrm{rec}} \ll \Gamma_{\mathrm{ph}}(\omega_0)$, which translates into

$$x_{\mathrm{qp}} \ll (\omega_0/\Delta)^{7/2}. \tag{30}$$

For typical values $\omega_0 \gtrsim 5\,\mathrm{GHz}$, $\Delta \sim 50\,\mathrm{GHz}$, and $x_{\mathrm{qp}} \lesssim 10^{-5}$, this condition is satisfied. For quasiparticles near the gap edge, the relevant process is that of photon absorption with rate $\bar{n}\Gamma_0$, see the last term in Eq. (2); then the condition $\bar{n}\Gamma_0 \gg \Gamma_{\mathrm{rec}}$ sets a lower bound on $\bar{n}$. For qubits with small aluminum islands, assuming at most a few quasiparticle in the islands we find $x_{\mathrm{qp}} \sim 10^{-5}$ and hence $\bar{n} > 10^{-2}$, which is typically satisfied, as discussed in the main text; of course if there is only one quasiparticle in an island, recombination cannot take place, and there is evidence that in systems with several islands the average number of quasiparticles in each island is less than one [10] or it can be suppressed to less than one [38]. For large volume qubits, even assuming $x_{\mathrm{qp}} \sim 10^{-8}$, we would find the requirement $\bar{n} > 1$, which cannot be met. On one hand, this means that in large qubits we cannot neglect the effect of generation/recombination in determining the energy dependence of the distribution function; on the other hand, since Eq. (30) is satisfied, Eq. (8) is still valid and our considerations about transition rates are unaffected.

## B  Solution to Eq. (14)

As discussed in the main text, at $x \gg x_{**} \approx \min\{r^{2/7}, r^{1/3}\}$, the asymptotic solution to Eq. (14) takes the form $F_r(x) \sim 1/\sqrt{x}$ for any $r$. That equation can be solved numerically by iteration; that is, starting with a trial function with the correct asymptotic behavior, we calculate the $(n+1)$th iteration by inserting the $n$th iteration into the right hand side of Eq. (14). Numerically, the integral is evaluated by splitting it into a "low" energy region, extending from $x$ to about $10 x_{**}$, and a high energy one; the contribution of the low-energy region is then obtained

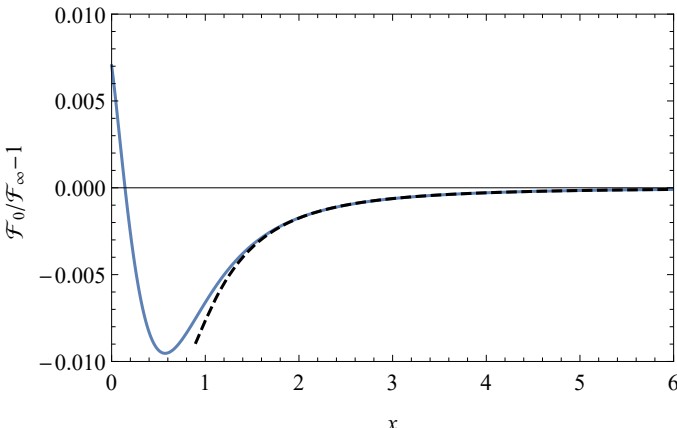

Figure 4: Solid line: deviation of the ratio $\mathcal{F}_0/\mathcal{F}_\infty$ from unity as obtained from numerical solutions to Eqs. (31) and (32). The dashed line is the analytic prediction given by the next-to-leading asymptotic terms [see text after Eq. (32)]; the agreement between the two curves at large $x$ further validates our numerical procedure.

by discretizing the integral on an equally-spaced grid with steps of order $0.02 x_{**}$, while the high-energy part is estimated analytically by using the leading asymptotic form of the solution. The calculation is stopped when the desired accuracy is reached; typically, the maximum relative change in the numerical solution becomes less than $10^{-6}$ after a small number ($\lesssim 8$) of iterations. We have checked that the solution thus found is only weakly sensitive (relative deviations at most of order $10^{-3}$) to $e.g.$ doubling the value of the splitting point of the integral or the resolution. The exact form of the initial trial function is unimportant: as long as we set it proportional to $1/\sqrt{x}$ for $x > x_{**}$, it can be represented for $x < x_{**}$ by an array of random numbers between 0 and 1. Interestingly, as we show next, the dependence of $\mathcal{F}_r$ on $r$ is, to a good degree, accounted for with an appropriate rescaling.

Let us consider the limiting cases $r \to 0$ and $r \to \infty$; by further rescaling variables by $r^{1/3}$ and $r^{2/7}$, respectively, we find the equations

$$\mathcal{F}_0(x) = \frac{105}{128} \int_x^\infty \frac{(x+y)(x-y)^2}{y^3 + 1} \frac{\mathcal{F}_0(y)\,dy}{y} \tag{31}$$

and

$$\mathcal{F}_\infty(x) = \frac{105}{128} \int_x^\infty \frac{(x+y)(x-y)^2}{y^3 + 1/\sqrt{y}} \frac{\mathcal{F}_\infty(y)\,dy}{y}. \tag{32}$$

Their asymptotic solutions at large $x$ are, up to an overall coefficient, $\mathcal{F}_0(x) \propto 1/\sqrt{x}(1 - 25/858x^3 + \ldots)$ and $\mathcal{F}_\infty(x) \propto 1/\sqrt{x}(1 - 11/512x^{7/2} + \ldots)$, respectively. At arbitrary $x$, we solve these equation numerically, as explained at the beginning of this section, and normalize the results so that $\mathcal{F}_0(x)/\mathcal{F}_\infty(x) \to 1$ as $x \to \infty$. Then the ratio between the two functions deviates from unity by less than 1 % at all $x$, as shown in Fig. 4. This result suggests that it should be possible to express $\mathcal{F}_r$ for any $r$ in terms of a single function with good accuracy. Indeed, let us define the average function $\bar{\mathcal{F}}(x) = [\mathcal{F}_0(x) + \mathcal{F}_\infty(x)]/2$, and normalize it so that $\bar{\mathcal{F}}(0) = 1$; our numerical result for this function is shown in Fig. 5 and it is accurately fit (within 0.1 %) by the Padé-like expression

$$\bar{\mathcal{F}}(x) \approx \sqrt{\frac{1 + 1.292x + 1.811x^2}{1 + 2.271x + 3.786x^2 + 5.155x^3}}. \tag{33}$$

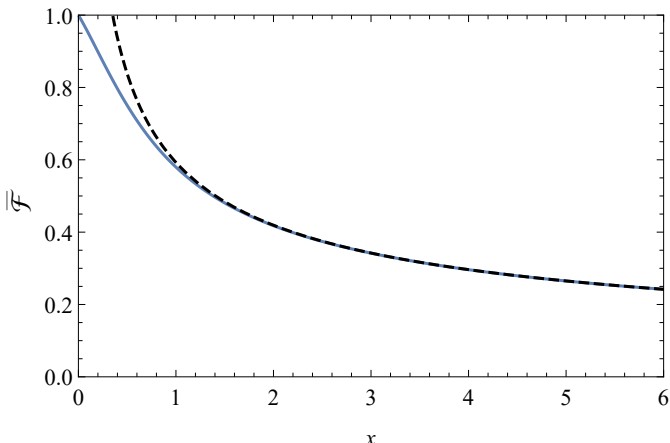

Figure 5: Solid line: $\bar{\mathcal{F}} = (\mathcal{F}_0 + \mathcal{F}_\infty)/2$ as obtained from numerical solutions to Eqs. (31) and (32); on this scale, the approximate formula in Eq. (33) is indistinguishable from the numerics. Dashed line: the asymptotic formula $0.5927/\sqrt{x}$, as obtained from Eq. (33) at large $x$, is a good approximation for $x > 1$.

For arbitrary value of $r$, we then find (within $\sim 1\%$)

$$\mathcal{F}_r(x) \approx \bar{\mathcal{F}}(\alpha(r)x)\mathcal{F}_r(0)\,, \tag{34}$$

with

$$\alpha(r) \approx 1/r^{1/3} + 1/r^{2/7} - 0.7345/r^{1/6+1/7}\,. \tag{35}$$

Here the power of the last term in the right hand side is arbitrarily set as the average between the powers of the first two, asymptotic terms, and the numerical factor is obtained by comparison with numerical solutions in the range $r$ from $10^{-3}$ to $10^3$. A more precise definition of the energy scale $T_{**}$ of Eq. (11) can be given as

$$\frac{T_{**}}{\omega_0} = \left(\frac{e^{\omega_0/T_0}}{\Lambda}\right)^2 \frac{1}{\alpha(r)}\,. \tag{36}$$

The above considerations were based on neglecting the second term in Eq. (10) in comparison to the first. To check this assumption, we note that using Eq. (12), at $\epsilon \gg T_{**}$ using the definitions in Eq. (13) we estimate the first term to be $(e^{\omega_0/T_0}\mathcal{F}_r(0)/r\sqrt{\alpha(r)}\Lambda^2)\sqrt{\omega_0/\epsilon}$. For the second term, using again those definitions and introducing the change of variable $x = t^2/\alpha(r)$, we find the approximate upper bound $2(e^{\omega_0/T_0}\mathcal{F}_r(0)/r\sqrt{\alpha(r)}\Lambda^2)$. Therefore we can indeed neglect the second term when $\epsilon \ll \omega_0$.

## C  Effect of finite phonon temperature

Most of the arguments given in this paper lead to finite results when the phonon temperature $T_{\text{ph}} \to 0$, so that only phonon emission is allowed. However, a problem arises when one tries to normalize the distribution function in Eq. (13) according to Eq. (25): the integral diverges logarithmically at low energies. This divergence should be cut off at $\epsilon \sim T_{\text{ph}} \ll T_{**}$:

$$x_{\text{qp}} = \int_{T_{\text{ph}}}^{\infty} \frac{d\epsilon}{\Delta}\sqrt{\frac{2\Delta}{\epsilon}} f(\epsilon) \sim \mathcal{F}(0)\sqrt{\frac{2\omega_0}{\Delta}} \frac{e^{5\omega_0/T_0}}{\Lambda^5}\left[e^{\omega_0/T_0}\ln\frac{T_{**}}{T_{\text{ph}}} + \Lambda\sqrt{\frac{T_{**}}{\omega_0}}\right]. \tag{37}$$

Let us check what happens if $T_{\rm ph}$ is not the smallest scale. If $T_{**} \lesssim T_{\rm ph} \ll \omega_0$, Eq. (8) remains valid. Thus, while at energies $\epsilon \sim T_{\rm ph}$ the in-scattering part of the collision integral is dominated by the phonon absorption, resulting in the thermal distribution $f(\epsilon) = f_0\, e^{-\epsilon/T_{\rm ph}}$, at energies $T_{\rm ph} \ll \epsilon \ll \omega_0$ we still have Eq. (12) and the distribution has therefore the power-law form $f(\epsilon) = f_\infty/\epsilon^4$. To relate the normalization constants $f_0$ and $f_\infty$ and estimate the crossover energy $\tilde{\epsilon}$ between exponential and power-law behavior, we note that the net probability current from small $\epsilon$ to $\epsilon > \omega_0$ due to absorption/emission of a quantum $\omega_0$ is given by

$$
\mathcal{J} = \int\limits_0^{\tilde{\epsilon}} \frac{d\epsilon}{\Delta}\sqrt{\frac{2\Delta}{\epsilon}}\, f_0\, e^{-\epsilon/T_{\rm ph}}\, \frac{\bar{n}\Gamma_0\Lambda\sqrt{\epsilon/\omega_0}}{e^{\omega_0/T_0} + \Lambda\sqrt{\epsilon/\omega_0}} \sim
$$
$$
\sim \frac{f_0}{\Delta\tau_{\rm ph}}\frac{T_{\rm ph}(\omega_0/\Delta)^3}{e^{\omega_0/T_0} + (\bar{n}\Gamma_0\tau_{\rm ph})^{-1}\omega_0^3\sqrt{T_{\rm ph}/\Delta^7}}, \tag{38}
$$

where in the last estimate we assumed that $\tilde{\epsilon}$ is at least a few times larger than $T_{\rm ph}$, so that the upper integration limit can be replaced with infinity; this assumption will be verified below.

The same current should be carried by the distribution function $f(\epsilon) = f_\infty/\epsilon^4$ at $T_{\rm ph} \ll \epsilon \ll \omega_0$, which corresponds to the phonon emission:

$$
\mathcal{J} = \int\limits_0^{\tilde{\epsilon}} \frac{d\epsilon''}{\Delta}\sqrt{\frac{2\Delta}{\epsilon''}}\frac{105}{128}\int\limits_{\tilde{\epsilon}}^\infty \frac{d\epsilon'}{\tau_{\rm ph}}\frac{f_\infty}{(\epsilon')^4}\frac{(\epsilon'+\epsilon'')(\epsilon'-\epsilon'')^2}{\sqrt{\Delta^7\epsilon'}} = \frac{599}{105\sqrt{2}}\frac{f_\infty}{\tau_{\rm ph}\Delta^4}. \tag{39}
$$

Equating the two expressions for $\mathcal{J}$, we find $f_\infty/f_0$. Thus, the equilibrium distribution $f_0\, e^{-\epsilon/T_{\rm ph}}$ crosses over to the power-law tail $f_\infty/\epsilon^4$ at the energy $\tilde{\epsilon}$ determined by the following equation:

$$
\frac{T_{\rm ph}\omega_0^3}{e^{\omega_0/T_0} + (\bar{n}\Gamma_0\tau_{\rm ph})^{-1}\omega_0^3\sqrt{T_{\rm ph}/\Delta^7}} \sim \tilde{\epsilon}^4 e^{-\tilde{\epsilon}/T_{\rm ph}}. \tag{40}
$$

The right-hand side of this equation has a maximum larger than $T_{\rm ph}^4$ at $\tilde{\epsilon} = 4T_{\rm ph}$, while the left-hand side is smaller than $T_{\rm ph}^4$ by virtue of the condition $T_{\rm ph} \gtrsim T_{**}$. Therefore the crossover energy $\tilde{\epsilon}$ exceeds $4T_{\rm ph}$ by a logarithmic factor. On one hand, this verifies the assumption used in Eq. (38); on the other hand, the presence of the power-law tail requires $\tilde{\epsilon} \ll \omega_0$, which is a stronger condition than just $T_{\rm ph} \ll \omega_0$.

## D  Electron-phonon interaction in the diffusive regime

The results presented in the main text are based on Eq. (5), which is valid for a clean metal, where the electron elastic mean free path due to static impurities is longer than the mean free path due to the electron-phonon scattering. Let us consider the opposite (diffusive) limit, when $F(\omega)$ is proportional to a different power of $\omega$. Here we analyze the case $F(\omega) \propto \omega$, relevant for impurities with fixed positions [23] and show that the results are qualitatively similar. For impurities which move together with the phonon lattice deformation, $F(\omega) \propto \omega^3$ [24, 25]; this case is also briefly discussed in the end of this section.

We write the function $F(\omega)$ appearing in the electron-phonon collision integral, Eq. (4), in the form [23]

$$
F(\omega) = \beta\,\frac{\omega}{\omega_D}, \tag{41}
$$

where the dimensionless slope $\beta$ depends in general on the electronic mean free path and $\omega_D$ is the Debye frequency. Assuming again low phonon temperature, we find for the relaxation rate

$$\Gamma_{\mathrm{ph}}(\epsilon) = \frac{16\pi}{5} \frac{\beta \epsilon^{5/2}}{\omega_D \sqrt{2\Delta}} \equiv \frac{1}{\tau_{\mathrm{ph}}} \left(\frac{\epsilon}{\Delta}\right)^{5/2}, \tag{42}$$

where the last expression redefines the characteristic electron-phonon time $\tau_{\mathrm{ph}}$ in the diffusive regime [cf. Eq (6)]. The kinetic equation (neglecting phonon absorption) has now the form

$$\frac{\partial f}{\partial t} = \bar{n}\Gamma_0 \sqrt{\frac{\omega_0}{\epsilon - \omega_0}} \left[ f(\epsilon - \omega_0) - e^{\omega_0/T_0} f(\epsilon) \right] + \bar{n}\Gamma_0 \sqrt{\frac{\omega_0}{\epsilon + \omega_0}} \left[ e^{\omega_0/T_0} f(\epsilon + \omega_0) - f(\epsilon) \right] +$$

$$+ \frac{5}{8} \int_\epsilon^\infty \frac{f(\epsilon')}{\tau_{\mathrm{ph}}} \frac{(\epsilon' + \epsilon)(\epsilon' - \epsilon) d\epsilon'}{\sqrt{\Delta^5 \epsilon'}} - \left(\frac{\epsilon}{\Delta}\right)^{5/2} \frac{f(\epsilon)}{\tau_{\mathrm{ph}}}. \tag{43}$$

Proceeding as in the main text, we find that for cold quasiparticles Eq. (10) becomes

$$\int_\epsilon^{\omega_0} \frac{(\epsilon' + \epsilon)(\epsilon' - \epsilon) f(\epsilon') d\epsilon'}{\sqrt{\omega_0^5 \epsilon'}} + \int_0^{\omega_0} \frac{f(\epsilon') d\epsilon'/\omega_0}{e^{\omega_0/T_0} + \sqrt{\epsilon'/\omega_0}\,\Lambda} =$$

$$= \frac{8}{5} \left[ \frac{\sqrt{\epsilon/\omega_0}}{e^{\omega_0/T_0} + \sqrt{\epsilon/\omega_0}\,\Lambda} + \left(\frac{\epsilon}{\omega_0}\right)^{5/2} \right] f(\epsilon), \tag{44}$$

where $\Lambda$ is now defined as

$$\Lambda = \frac{(\omega_0/\Delta)^{5/2}}{\bar{n}\Gamma_0 \tau_{\mathrm{ph}}}. \tag{45}$$

The equation can be simplified for energies [cf. Eq. (11)]

$$\epsilon \gg T_{**} \sim \min\left\{ \Delta(\bar{n}\Gamma_0 \tilde{\tau}_{\mathrm{ph}})^{2/5}, \omega_0 e^{-\omega_0/(2T_0)} \right\}, \tag{46}$$

in which case it reduces to

$$\epsilon^{5/2} f(\epsilon) = \frac{5}{8} \int_\epsilon^\infty \frac{(\epsilon' + \epsilon)(\epsilon' - \epsilon) f(\epsilon') d\epsilon'}{\sqrt{\epsilon'}}. \tag{47}$$

The solution to this equation is $f(\epsilon) = C/\epsilon^3$: although the power of this tail is different from that found for the clean metal case, the qualitative behavior of the distribution function is the same. Next, we show that the qualitative similarities between clean and diffusive cases persist also for hot quasiparticles.

For hot quasiparticles we approximate again the photon collision integral via a diffusion operator, Eq. (15), and the kinetic equation becomes

$$\frac{\partial f}{\partial t} = \bar{n}\Gamma_0 \sqrt{\omega_0^5 \epsilon}\, \frac{\partial}{\partial \epsilon} \left\{ \frac{e^{-\epsilon/T_0}}{\epsilon} \frac{\partial}{\partial \epsilon} \left[ e^{\epsilon/T_0} f(\epsilon) \right] \right\}$$

$$+ \frac{5}{8} \int_\epsilon^\infty \frac{f(\epsilon')}{\tilde{\tau}_{\mathrm{ph}}} \frac{(\epsilon' + \epsilon)(\epsilon' - \epsilon) d\epsilon'}{\sqrt{\Delta^5 \epsilon'}} - \left(\frac{\epsilon}{\Delta}\right)^{5/2} \frac{f(\epsilon)}{\tilde{\tau}_{\mathrm{ph}}}. \tag{48}$$

The temperature scale $T_*$ should be now defined as $T_* = \omega_0 \Lambda^{-1/5}$, so for $T_* \ll T_0$, using dimensionless variables, we obtain the steady-state equation as

$$0 = \sqrt{x}\, \partial_x \left[ x^{-1} \partial_x f(x) \right] - x^{5/2} f(x) + \frac{5}{8} \int_x^\infty \frac{dy}{\sqrt{y}} (y + x)(y - x) f(y). \tag{49}$$

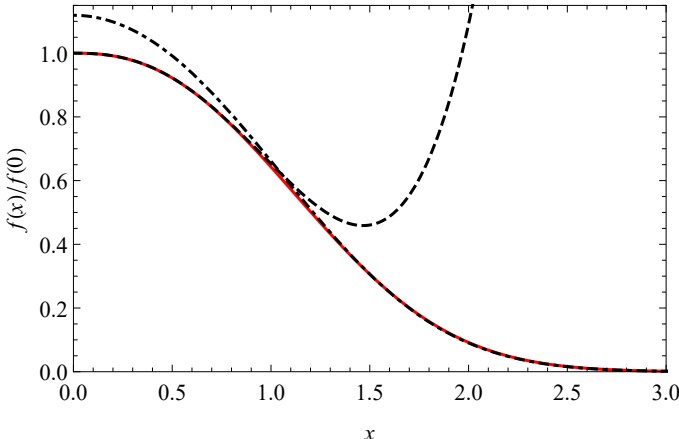

Figure 6: Red solid line: numerical solution to Eq. (49). Black dashed: low energy approximation, Eq. (53). Black dot-dashed: high energy approximation, Eq. (51).

At high energies, $x \gg 1$ (but still $x \ll T_0/T_*$), we can neglect the last term, and with the change of variable $x = 4^{1/5}\sqrt{z}$ we arrive at the generalized Airy equation

$$0 = \partial_z^2 f - z^{1/2} f \ , \tag{50}$$

whose solution can be written in terms of the modified Bessel function of the second kind to give

$$f(x \gg 1) \approx f_\infty x K_{2/5}\left(\frac{\sqrt{4}}{5} x^{5/2}\right), \tag{51}$$

with asymptotic behavior

$$f(x \gg 1) \sim f_\infty \sqrt{\frac{5\pi}{2\sqrt{4x}}} e^{-\frac{\sqrt{4}}{5} x^{5/2}}. \tag{52}$$

The solution for $x \to 0$ can be found in the form of a power series [cf. Eq. (20)]

$$f(x) \approx f_0\left(1 - a x^{5/2} + b x^{9/2} + \dots\right), \tag{53}$$

where

$$a = \frac{1}{2}\int_0^\infty dy\, y^{3/2}\frac{f(y)}{f_0}\,, \quad b = \frac{1}{18}\int_0^\infty dy\, y^{-1/2}\frac{f(y)}{f_0}\ . \tag{54}$$

Note that proper normalization can be found if $b$ is known. As in Sec 3.2, using the numerical solution to Eq. (49) in these definitions we find $a \approx 0.468$ and $a \approx 0.121$. Fitting the numerical solution we also obtain $f_\infty \approx 0.53 f_0$. The numerical solution and the analytical approximations are plotted in Fig. 6.

In the regime $T_* \gg T_0$, we find again that the distribution function takes the Boltzmann form up to high energies:

$$f(\epsilon) \propto \begin{cases} e^{-\epsilon/T_0}, & \epsilon \ll T_\infty, \\ e^{-\epsilon/2T_0 - \sqrt{3}\epsilon^2/4T_0 T_\infty}, & |\epsilon - T_\infty| \ll T_\infty, \\ e^{-\epsilon/2T_0}\frac{\epsilon}{2T_0} K_{2/5}\left[\frac{2}{5}\left(\frac{\epsilon}{T_*}\right)^{\frac{5}{2}}\right], & \epsilon \gg T_\infty, \end{cases} \tag{55}$$

where $T_\infty = 2^{1/3} T_* (T_*/2T_0)^{2/3}$.

Finally, we briefly discuss the case $F(\omega) \propto \omega^3$, which corresponds to the cooling power in the normal state proportional to $T^6 - T_{\text{ph}}^6$, also observed in experiments [39]. All steps are fully analogous. Writing $F(\omega) = \beta(\omega/\omega_D)^3$, we obtain the relaxation rate

$$\Gamma_{\text{ph}}(\epsilon) = \frac{128\pi}{63} \frac{\beta \epsilon^{9/2}}{\omega_D^3 \sqrt{2\Delta}}. \tag{56}$$

The numerical coefficient in front of the integral in $\text{St}_n f(\epsilon)$ is $63/64$. In the cold quasiparticle regime, the power-law solution at

$$\epsilon \gg T_{**} \sim \min\left\{\Delta(\bar{n}\Gamma_0\tilde{\tau}_{\text{ph}})^{2/9}, \omega_0 e^{-\omega_0/(4T_0)}\right\}$$

is $f(\epsilon) \propto 1/\epsilon^5$. Here $\Lambda$ and $T_*$ should be defined as

$$\Lambda = \frac{(\omega_0/\Delta)^{9/2}}{\bar{n}\Gamma_0\tau_{\text{ph}}}, \quad T_* = \omega_0 \Lambda^{-1/7}.$$

In the hot regime, the generalized Airy equation obtained at high energies after the substitution $x = 4^{1/5}\sqrt{z}$ is

$$0 = \partial_z^2 f - z^{3/2} f, \tag{57}$$

whose solution is expressed in terms of the modified Bessel function $K_{2/7}$.

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
