# Peer review of "Non-equilibrium quasiparticles in superconducting circuits: photons vs. phonons"

_SciPost Physics, doi:SciPost Phys. 6, 013 (2019)_

## Round 1 · Referee Report · Anonymous (Referee 1) · 2018-12-8

Strengths

1- detailed analytical and numerical study of an intricate non-equilibrium kinetic problem in the presence of two baths.

2- clear discussion on the relevance of the studied model to recent experiments on the out-of-equilibrium quasiparticle distribution in superconducting qubits.

Weaknesses

1- difficulty to grasps the main findings due to the large number of studied regimes (see requested changes)

Report

The authors study theoretically the energy dependence of the distribution function of superconducting quasiparticles coupled with a phonon bath and a low-energy (on the scale of the superconducting gap) driven qubit (or resonator mode). They find a variety of behaviors, depending on the qubit (or resonator) frequency and its effective temperature, the temperature of the phonon bath, and the ratio between the parameters that characterize the coupling of the quasiparticles to phonons and to the qubit (or resonator). They apply their results to the estimation of the quasiparticle contribution to the qubit relaxation and excitation, which have been recently measured in transmon qubits. They conclude that a qubit by itself cannot overheat quasiparticles. This points out to the necessity of adding additional mechanisms to the ones considered in the present study if one aims at understanding some recent experiments.

Overall, I find that the results are interesting and timely, and I recommend the manuscript to be published in Sci-Post, provided the author consider the following requested changes.

Requested changes

1- Plots indicating qualitatively the energy dependence of the distribution function in various energy windows, both in the so-called cold and hot quasiparticle regimes, would be helpful for the reader to grasp the main findings;

2- In Sec. 3.2, it is not particularly clear that the results are first summarized before they are derived (which looks like a different choice from the one made for the presentation of the results in Sec. 3.1);

3- It is not particularly obvious where the equation given in the footnote 1 (especially the numerical factors $\sqrt{3}$ and $\sqrt{128}$ inside it) comes from.

4- Eq. (7) seems to assume that the first term in the rhs of the non-numbered equation below Eq. (6) can be neglected in front of the second one. Is it obvious, or should it be checked self-consistently after a solution is found?

---

## Round 2 · Referee Report · Anonymous (Referee 1) · 2019-1-7

Report

The authors adequately responded to the comments in my first report. I now fully recommend the publication of their manuscript in SciPost.

---

## Round 2 · Author Response

Dear Editor,
we submit a revised version of our manuscript. We have taken into account all the Referee's comments, so we are hopeful that the paper can be accepted for publication in its present form.

---

## Round 2 · List of Changes

We thank the referee for the comments. We agree with them, here are the details of the modifications.

  • We added a figure in the third section, where we show schematically the distribution function in the two regimes.

  • In Sec. 3.2, we moved the summary of its results to the end of the section.

  • We gave a label to the last equation of Sec. 2 and expanded the discussion in the beginning of Sec. 3.1, to address the last remark of the referee.

  • We added the text to the footnote, to clarify how the equation is obtained.

---

## Editorial Decision

published